# OpenReview forum: "Noise Injection Reveals Hidden Capabilities of Sandbagging Language Models"
_NeurIPS.cc/2025/Conference — NeurIPS 2025 poster_

### Official Review · Reviewer_PghU · 2025-06-30

**Clarity:** 4
**Significance:** 3
**Originality:** 3
**Rating:** 4
**Confidence:** 3

**Summary:**

LLMs can strategically under perform in capability assessments provided by regularity bodies when they are explicitly directed or even when they are implicitly motivated in pursuit of a higher goal.  To evaluate LLMs accurately, methods which can detect sandbagging is needed. Existing sandbagging detection methods are grouped into two: i. capability elicitation and ii. deception detection. Capability elicitation methods focus on fine-tuning the LLMs to reveal their unhindered performance while deception detection methods instead rely on probing the behavior of the model (e.g. its reasoning) or modifying the internal states of a model (e.g. its activation units).

The submitted paper propose a novel deception detection method that relies on adding noise to the weights of an LLM. The underlying idea is noise injection will make it harder for the LLMs to comply with sandbagging, and, hence, adding the right amount of noise can lead to an increased performance in the sandbagged LLM. Accordingly, proposed method inserts additive Gaussian noise to a tested LLM's weights and evaluates the performance at different noise variances in a particular benchmark dataset. Next, they calculate the "improvement ratio '', i.e. the ratio of the best performance achieved under noise injection (grid-searched over different seeds and noise variances) to the performance achieved without noise injection and feed it as an input feature to a decision tree-based classifier for predicting whether the LLM is sandbagged or not.

In their experiments authors test their approach considering different sandbagging techniques, LLM models and datasets. The authors show that the improvement ratio is statistically significant for sandbagging LLMs and non-sandbagging LLMs. Authors also report, the classifier that inputs the improvement ratio achieves 96.2\% accuracy in leave-one-model-out setting and is highly successful in detection the sandbagging.

**Questions:**

-Can you please clarify which entries correspond to  the no-goal-setting in Table 1?

-It is stated, ``we calculate 95\% confidence intervals with the Benjamini-Hochberg correction [8] at $\alpha = 0.05$ and conduct a two-tailed t-test, assuming unequal variances.'' Can you clarify whether the BH correction is applied to the p-values or the confidence intervals?

-In Table 1, have you applied the BH correction across different rows? In each row how many observation did you used for computing the p-values?

-Line 131: what does $N(\mu)$ refers to?

-Would it be possible for you to report the model-specific accuracy in the leave-one-model-out setting?

-In your classification setting, did you use any scaling or standardization for the ``improvement ratios'' while combining data across different models and have you tested using a linear classifier?

-How does quantization effects your detection performance, would you expect detection accuracy to be higher, or, lower for the same model under a lower quantization level?

-How was the sigma range and step size selected?

**Ethical Concerns:**

["NO or VERY MINOR ethics concerns only"]

**Final Justification:**

keeping the original score. A better version is possible which is not a borderline paper.

**Limitations:**

yes

**Quality:**

3

**Strengths And Weaknesses:**

Quality:
-The authors propose a technically sound method that follows a different take on the idea of [15] ``Intoxicate models so they reveal bad behavior
perturbing''. [15] uses perturbing the intermediate hidden representations while the submission perturbs the model weights.

-The experiments authors conducted (in the main text) all show their method is highly successful in detecting sandbagging. It is important to characterize the setting in which the proposed method faces a difficulty, e.g. it is mentioned some difficulty with ``phi-3-small'' without further explanation. I was hoping to authors can give specific examples where their approach produces false positives.

-The experiment section can be improvement with a baseline detection method. The authors state this is due to existing methods limitations (e.g. requiring a fine-tuning dataset). I am not familiar with the literature to judge this but the method in [15] seems like can be tweaked to be tested in ``improvement ratio'' setting.

Clarity:
-The paper is easy to follow and engaging.

Significance:
-The authors propose a practically applicable method in the gray-box setting where LLM weights can be updated. Authors contribute to collective understanding of the sandbagging behavior of LLMs by demonstrating it can be turned off to an extend by noise injection to LLM weights.

Originality:
-I am not completely familiar with the literature to judge its originality. Compared to [15] which added noise to activation of LLMs, this submission uses a different approach adding noise to model weights and provide a more complete investigation of its effectiveness.

---

> ### Author Rebuttal · Authors · 2025-07-31
>
> Reviewer PghU,
>
>
> We greatly appreciate your review, and have found your feedback helpful in improving our work. We are pleased that the reviewer agrees our method is both technically sound, and highly successful. We found the provided critiques valuable, and have addressed each explicitly below.
>
> ---
>
> **It is important to characterize the setting in which the proposed method faces a difficulty…**
>
> ---
>
> Out of all 95 runs of our XGBoost classifier, we only saw four false positives:
> | Model | Dataset | $\phi$ |
> |---|---|---|
> | Llama-8b | GSM8k | 1.235 |
> | Llama-8b | GSM8k | 1.294 |
> | Llama-8b | GSM8k | 1.279 |
> | Mistral-7B | MMLU | 1.118 |
>
> We expect that we observed multiple false positives for the Llama-8B run of the GSM8k dataset due to the limited number (30) of questions sampled from the dataset. This number was selected due to the high time-allocation on GPUs; running the entire experiment for 30 questions from GSM8k took 3 days to complete.
>
> ---
>
> **The experiment section can be improved with a baseline detection method… Not completely familiar with the literature to judge its originality… this submission uses a different approach of adding noise to model weights and provides a more complete investigation of its effectiveness.**
>
> ---
> We thank the reviewer for highlighting this important comparison with [15] (Clymer et al., 2024).
> We agree with the reviewer that our work provides a more complete investigation through systematic parameter sweeps and rigorous statistical testing. We want to point out that beyond this, our approach differs meaningfully in two key ways:
>
> 1. **Practical Deployment.** Over the last week, we attempted to implement POSER's activation noise injection method as a baseline but encountered significant implementation challenges. POSER requires modifying the model's forward pass to inject noise into intermediate activations at specific layers. This approach is difficult to integrate with existing inference frameworks, with vLLM choosing not to implement the feature. In contrast, our weight noise injection method using LoRA adapters is compatible with practically all modern inference frameworks like vLLM, making it more accessible for practitioners and easier to deploy at scale.
>
> 2. **Broader applicability**: While POSER was evaluated on a narrow task (binary detection of alignment faking in 324 role-playing model pairs), our method demonstrates effectiveness across diverse settings: multiple benchmarks (MMLU, AI2, WMDP, GSM8k), various model architectures, and crucially, an in-the-wild sandbagging case (Mistral Large) without requiring paired models or synthetic scenarios.
>
> We will add this comparison with POSER in Section 6 (Related Work) to better contextualize these contributions.
>
> ---
>
> **Can you please clarify which entries correspond to the no-goal-setting in Table 1?**
>
> ---
>
> In the version of Table 1 in our original submission, no entries correspond to the no-goal-sandbagging setting, as these results are only discussed in the text of the submission. We plan to add a clearly labeled row to Table 1 with the measurements seen in  this setting.
>
> ---
>
> **It is stated, ``we calculate 95% confidence intervals… Can you clarify whether the BH correction is applied to the p-values or the confidence intervals?**
>
> ---
>
> This sentence is poorly stated on our end, and we appreciate you flagging this. To clarify, **we apply the BH correction to the p-values.** We have adjusted wording in our manuscript to reflect this.
>
> ---
>
> **In Table 1, have you applied the BH correction across different rows?…**
>
> ---
>
> Yes, the BH correction was applied to the p-values across the different rows in Table 1.
>
> For each row of Table 1, we conduct 5 experimental runs (observations) for both the standard and sandbagging model variants, each with $N=5$ noise injection trials (L134-140).
>
> ---
>
> **Line 131: what does N(mu) refers to?**
>
> ---
>
> This was an artifact from an older version of our manuscript, and a typo. We have replaced $\mathcal{N}(\mu)$ with the symbol for a single noise vector, $\epsilon$.
>
> ---
>
> **Would it be possible for you to report the model-specific accuracy in the leave-one-model-out setting?...**
>
> ---
>
> We thank the reviewer for the suggestion, and have added this to an appendix. As you can see, we retrieve the lowest accuracy with Llama-8b. This is primarily due to the false positives in the math setting with a subset of only 30 questions.
>
> Note that, for each model/benchmark pair (row) in Table 1 we have 10 observations (5 standard + 5 sandbagging) on which to evaluate our XGBoost classifier, except for Llama-70b pairings, each of which has only 2 observations (1 standard + 1 sandbag) due to compute costs. For a given model/benchmark pair, we train the classifier on all labeled observations which do not include the model being assessed, including the newly added runs for Mistral-7B-Finetuned and Deepseek.
>
> Model-Specific Accuracy (Leave-One-Model-Out CV):
> | Model | Accuracy | Number of Samples |
> |---|---|---|
> | Llama-8b | 0.90 | 50 |
> | Mistral-7b | 0.98 | 40 |
> | Phi3-medium| 1.0 | 30 |
> | Phi3-mini | 0.93 | 30 |
> | Phi3-small | 0.90 | 30 |
> | Deepseek | 1.0 | 10 |
> | Mistral-7b-finetuned | 1.0 | 10 |
> | Llama-70b | 1.0 | 6 |
>
> ---
>
> **In your classification setting, did you use any scaling or standardization for the ``improvement ratios''...**
>
> ---
>
> By construction, the improvement rations $\phi$ are normalized. This is because $\phi$ is defined as the maximum performance seen across $N$ noise injection trials divided by the performance seen when no noise is applied to the model (i.e. the noise vector is $\vec{0}$).
>
> We have tested using a linear classifier instead of the XGBoost decision tree which we went with for the paper. The following results are for a linear classifier trained on the same data as the classifier we reported in our work:
>
> Model-Specific Accuracy (Leave-One-Model-Out CV)
> | Model | Accuracy | Number of Samples |
> |---|---|---|
> | Llama-8b | 0.76 | 50 |
> | Mistral-7b | 0.98 | 40 |
> | Phi3-medium| 1.0 | 30 |
> | Phi3-mini | 0.73 | 30 |
> | Phi3-small | 0.77 | 30 |
> | Deepseek | 1.0 | 10 |
> | Mistral-7b-finetuned | 1.0 | 10 |
> | Llama-70b | 1.0 | 6 |
>
> Mean accuracy across folds: 0.87
> Weighted F1-score across folds: 0.85
>
> ---
>
> **How does quantization effects your detection performance,…**
>
> ---
>
> It’s fairly unclear how quantization would affect the results of noise injection. A priori we expect this to have a marginal impact, since quantization wouldn’t change how information or preferences are stored within a model.
>
> However, one could consider quantization to be a form of noise injection itself, since decreasing the precision of the weights may have the effect of running a single noise vector $\epsilon$ across the model in a magnitude proportional to the amount of quantization. If after quantization, a strong performance boost was observed on a dangerous capability benchmark (after worries about just scrubbing away safety training were addressed) I think this would be a fairly high cause for concern.
>
> ---
>
> **How was the sigma range and step size selected?**
>
> ---
>
> This specific range, $\sigma \in [0, 0.01]$ was selected because we observed that model performance drops to around 0 $\forall \sigma > 0.01$.
> The step size $\Delta\sigma=0.001$ was selected through iteration on small-scale versions of our experiment. We found that this step size was a balance between granularity of our analysis, and the practical concerns of compute time and cost. In practice, it is possible that one could select a larger step size for quick feedback, but we do not investigate the relationship between step size and accuracy of our approach in this work.

---

> > ### Comment · Reviewer_PghU · 2025-08-05
> >
> > The authors addressed some of the comments and presented extra work to clarify the results however the paper would benefit from a major revision thus I will hold my initial rating.

---

### Official Review · Reviewer_mYUM · 2025-07-03

**Clarity:** 3
**Significance:** 3
**Originality:** 3
**Rating:** 4
**Confidence:** 4

**Summary:**

The paper is focusing on the problem of accurately estimating the capabilities of AI systems. The main issue with such evaluations is when models strategically underperform in such evaluations (either due to explicit developer intervention or unintended model behavior), a phenomenon known as sandbagging. To detect sandbagging, the authors propose injecting noise into model weights, which leads to models exhibiting anomalous performance improvements. The underlying hypothesis behind the method is that injecting noise into a model exhibiting its true capabilities should never substantially improve performance; anomalous increases in performance means that sandbagging behavior is being disrupted. Noise injection is performed either through direct parameter modification or noise application through LORA adapters which only modifies the linear layer parameters of the model. The authors conduct experiments across various model sizes, model families, and sandbagging methods (prompted sandbagging, password-locked models and no-goal setting) and find that sandbagging models frequently demonstrate significant performance improvements before eventual degradation at higher noise levels (in contrast to standard models that show expected degradation pattern with increasing noise).

**Questions:**

How many noise injection trials are required to reliably establish an LLM model is sandbagging? Why conduct N=5 noise injection trials? Have you tried values greater than this?

While the method works in practice, what is the likelihood that the added noise will disrupt sandbagging behavior for a given LLM model?


Line 199: typo - reproducibility

**Ethical Concerns:**

["NO or VERY MINOR ethics concerns only"]

**Final Justification:**

Thanks for the authors for their response. I will keep my score.

**Limitations:**

yes

**Quality:**

3

**Strengths And Weaknesses:**

Strengths:

The paper addresses an important and relevant topic, i.e. accurately assessing the capabilities of LLM models. The field is in need of such methods for accurately monitoring progress in LLM models and for ensuring their safety and oversight.

The proposed method is intuitive, simple, elegant and model-agnostic. The paper makes important contributions showing that injecting noise into sandbagging LLMs can significantly increase their performance, allows to distinguish sandbagging models from standard ones across various model architectures, sizes, and datasets, and finds models such as Mistral Large 120B exhibiting sandbagging without explicit instruction to do so.

Compared to linear probes and activation based detection methods, noise injection not only identifies the presence of sandbagging but also generates concrete examples of improved performance. Besides, the method is unsupervised and can recover maximum accuracy in some settings.

Experiments on multiple benchmarks (WMDP, AI2, MMLU, GSM8K and BoolQ) support the authors claims.


Weaknesses:

The authors claim their method is completely unsupervised, however in the experiments they train an XGBoost decision tree classifier to distinguish between standard models and sandbagging models. It would be good for the authors to clarify why this is needed.

Lack of theoretical guarantees that the added noise will disrupt sandbagging behavior

---

> ### Author Rebuttal · Authors · 2025-07-31
>
> Reviewer mYUM,
>
> We thank you for your review, and have found your insights valuable for improving our paper. We are encouraged that the reviewer found noise injection intuitive, elegant, and model agnostic, as these were key priorities for our team. The critiques provided were all useful for improving our paper, and we have explicitly addressed each below.
>
> ---
>
> **Authors claim method is completely unsupervised… however they train an XGBoost decision tree classifier…**
>
> ---
>
>
> To clarify, the validity of noise injection can be confirmed by conducting multiple experimental runs and calculating p-values; this is the method used to obtain the p-values in Table 1.
>
> Our reason for including the classifier is to provide a method which can classify a *single* experimental run without oracle knowledge of the subject models capabilities. This substantially decreases the cost of our approach; this is particularly relevant for larger models, which could be prohibitively expensive to work with. The classifier provides a tool that can be assessed for the efficacy of our method in this more realistic use case.
>
> However, given that the largest improvement ratio observed in standard models across all model/benchmark pairs was 1.294 (Llama-8b on GSM8k), and the improvement ratios for a vast majority of sandbagging models were greater than this value, simply ‘eyeballing’ a plot similar to Figure 2 (but with only one model) is sufficient in most cases!
>
> ---
>
> **Lack of theoretical guarantees…**
>
> ---
>
> We agree that further work understanding noise injection in a more theoretical manner is indeed interesting, but we see this as out-of-scope for this work.
>
> ---
>
> **How many noise injection trials are required to reliably establish an LLM model is sandbagging? Why conduct N=5 noise injection trials?**
>
> ---
>
> To validate our decision of $N=5$ noise injection trials we have included analysis for $N \in [1,5]$ (no bootstrapping, we just take the lowest $N$ numeric seeds). For each $N$ we ran the same XGBoost classifier used in our primary analysis, and also included a linear classifier. Note that the values for $N=5$ in this table don’t perfectly match those seen in Table 1 of our submission, as we conducted additional experimental runs to calculate p-values not originally present.
>
> | $N$ | Total comparisons | Significant (BH) | XGBoost Accuracy | Linear Accuracy |
> |-----|-----|-----|-----|-----|
> | 5 | 19 | 19 | 0.9611 | 0.8669 |
> | 4 | 19 | 19 | 0.9440 | 0.8789 |
> | 3 | 19 | 18 | 0.9219 | 0.8776 |
> | 2 | 19 | 18 | 0.9029 | 0.8814 |
> | 1 | 19 | 18 | 0.8460 | 0.8693 |
>
> We chose to report values for $N=5$ seeds, because it is the smallest value for which our classifier achieves an accuracy greater than 95%.
>
> In addition, we agree that understanding the relationship between number of seeds sampled and performance is key to our work. Our exploration of this relationship is visualized in Figure 3, which plots the maximum percentage of baseline performance recovered from $N \in [1,100]$ noise injection trials. Based on feedback from multiple reviewers, we realize that Figure 3 was not clear in what it visualized. Although we cannot demonstrate the modifications to our figure during this stage, as links and PDF rebuttals are not allowed this year, we have made substantial modifications to the figure.
>
> ---
>
> **While the method works in practice, what is the likelihood that the added noise will disrupt sandbagging behavior for a given LLM model?**
>
> ---
>
>
> This is a great idea which substantially adds to our analysis. Provided we are given an additional 10th page for the main conference paper, as has been the case in the past, we plan to include the results below in the main body of the paper; otherwise, we will add an appendix item to capture these results.
>
> For each model/benchmark pair, we let $\phi_{std,max}$ be the maximum increase in performance recorded across all noise injection trials. We then calculate the probability that the improvement ratio measured for an individual noise injection trial, $\phi\left(\eta\right)$, is greater than $\phi_{std,max}$.
>
> | Model | $ P\left(\phi\left(\mathcal{N}\left(\mu\right)\right) > \phi_{std,max}\right)$ |
> |-----|-----|
> | Llama-8b | 0.61 |
> | Mistral-7b | 0.81 |
> | Mistral-7b-finetuned | 1.00 |
> | Phi3-medium | 0.97 |
> | Phi3-mini | 0.96 |
> | Phi3-small | 0.79 |
>
> | Dataset | $ P\left(\phi\left(\mathcal{N}\left(\mu\right)\right) > \phi_{std,max}\right)$ |
> |-----|-----|
> | AI2 | 0.96 |
> | MMLU | 0.82 |
> | WMDP | 0.79 |
> | GSM8K | 0.58 |
> | Overall | 0.82 |

---

> ### Author Response · Authors · 2025-08-06
>
> Dear Reviewer mYUM,
>
> Thank you for your insightful review. We hope our responses have clarified the unsupervised nature of our method and addressed your questions about the number of noise injection trials. Please let us know if you need any further clarification.

---

### Official Review · Reviewer_QKfV · 2025-07-03

**Clarity:** 4
**Significance:** 3
**Originality:** 4
**Rating:** 5
**Confidence:** 3

**Summary:**

This paper introduces a noise-injection probe for detecting “sandbagging” intentional underperformance in large language models. By adding low-variance Gaussian noise to model weights or LoRA adapters, the authors find that sand-bagged models often exhibit notable performance gains, while standard models remain largely unaffected. They capture this effect using an improvement ratio,

\$\phi = \max(\text{noised accuracy}) / \text{baseline accuracy}\$.

The authors present the probe as a lightweight, model-agnostic auditing tool for developers and regulators. However, its generalizability and practical usability still require further validation.

**Questions:**

**Q1**. Can activation-space or prompt-space noise replicate the $\phi$ signal without weight access?

**Q2**. How does your probe compare to linear-probe detectors or fine-tune elicitation on the same tasks?

**Q3**. Have you tried coding/tool-use benchmarks such as BigCodeBench to test more fragile behaviours?

**Ethical Concerns:**

["NO or VERY MINOR ethics concerns only"]

**Limitations:**

Please see the weakness section.

**Quality:**

3

**Strengths And Weaknesses:**

### **Strengths**

- **Simple and lightweight method:** The noise-injection probe is easy to implement, requiring only a few lines of code and no additional data, benchmarks, or probes.
- **Wide empirical coverage:** The study spans seven model sizes (7B–120B) and evaluates performance across four diverse benchmarks (AI2, MMLU, WMDP, GSM-8K).
- **Robust and consistent signal:** All 17 prompted sand-bagging model pairs exhibit notable improvements in \$\phi\$; even a password-locked DeepSeek-7B shows a 5× gain on GSM-8K.
- **First real-world validation:** Demonstrates full recovery of Mistral Large’s performance in a no-goal setting, without relying on sabotage-specific prompts.

### **Weaknesses**

- **Limited scalability analysis:** The probe's behavior is inconsistent across model sizes, and it fails to address ceiling effects in already high-performing models where \$\phi\$ has little room to grow.
- **Statistical concerns:** Use of only five seeds per test and reliance on extreme-value reporting may inflate \$\phi\$ estimates; several key p-values are missing, particularly in Table 1.

---

> ### Author Rebuttal · Authors · 2025-07-31
>
> We thank the reviewer for their work in reviewing our paper!
>
> We appreciate that the reviewer agrees our method is validated on a wide range of benchmarks, and produces a robust signal for detecting sandbagging. We also appreciate your critiques, and have revised our paper to address each of them, including a comparison to similar work [15], which is detailed in the response to reviewer PghU.
>
>
> ---
>
> **Limited scalability analysis: The probes behavior is inconsistent…**
>
> ---
>
> We thank the reviewer for bringing up this point, and would like to clarify that our method does scale with model size. This is evidenced by the varying parameter count of models seen in Table 1, including three models of varying sizes from the same family (Phi3).
>
> We also want to clarify that what we describe as “scalability analysis” (Figure 3) in the paper refers to increasing the number of seeds sampled (i.e. number of noise injection trials  $N$), not model size. We have modified these mentions to instead use the label “seed sampling experiments,” and have updated the figure substantially. We would include an image, but cannot due to this year’s limitations on including links and PDF updates.
>
> ---
>
> **Statistical concerns: Use of only five seeds per test…**
>
> ---
> **On the use of five-seeds:**
> We agree that further examination of the number of seeds sampled is useful in  understanding the value of our work. In addition to experimentation described in Figure 3, which examines maximum performance obtained by a model for $N$ noise injection trials, we have added justification for our choice of $N=5$ seeds. These additions are discussed in detail in our response to Reviewer mYUM.
>
> **On the missing p-values:**
> Thank you for pointing this out. We’ve now run additional experiments for Mistral-7b† (fine-tuned) on WMDP and Llama-8b on BoolQ with 5 runs for both sandbagging and standard models to calculate p-values. Due to prohibitive costs, we did not conduct additional runs for Llama-70b.
>
> | Model | Standard | Sandbag | p-value |
> |---|---|---|---|
> | Mistral-7b† | 1.01 (0.01) | 3.12 (0.2) | 0.004 |
> | Llama-8b (BoolQ) | 1.05 (0.01) | 1.21 (0.02) | 0.0001 |
>
>
> These p-values remained significant after BH corrections as well. These results will be included in an updated version of the table for our full publication.
>
> ---
>
> **Q1. Can activation-space or prompt-space noise replicate the signal without weight access?**
>
> ---
>
> This is certainly an interesting question, as the validity of prompt-space noise would open up new avenues for investigation into closed-source models. We are aware of one effort which is actively investigating this area.
>
> Activation-space noise is similar to the approach by [15] (Clymer et al., 2024), which is described in detail in our response to reviewer PghU. In short, [15] showed activation noise injection can work in limited settings in toy role-playing scenarios (not sandbagging), though it requires custom modifications to the model's forward pass that are difficult to integrate with standard inference frameworks."
>
> ---
>
> **Q2. How does your probe compare to linear-probe detectors…**
>
> ---
>
> To address this concern, we look to Detecting Strategic Deception Using Linear Probes (arxiv:2502.03407) which investigated whether linear probes trained on toy data could identify anomalous underperformance (e.g. sandbagging) in more contexts dissimilar from those seen during training. They found that this method was quite robust; when setting the false-positive rate to 1%, the true-positive rate was 95-99%.
>
> However, this method only identifies sandbagging, whereas noise injection provides concrete examples of improved performance, as noted by reviewer mYUM. Additionally, their work only examined models prompted to sandbag, and our works investigated meaningfully different tasks. As such, there has not yet been an apples-to-apples comparison of noise-injection to performance of linear-probes. We believe this is an ideal area for future investigations, but is out of scope for this work.
>
> Other work (arxiv:2507.12691, added to arxiv on July 16, 2025) compares the efficacy of linear probes by investigating performance of an auditor with and without access to the results from the linear probes trained in the first study mentioned.
>
> As a final note, we expect the failure modes of these two methods, i.e. noise injection and linear probes, will not be identical; for example, it is possible that representations of sandbagging aren’t stored linearly within the model. This means that both methods could be used in conjunction to increase the likelihood of identifying unwanted behavior, as is discussed in our reference to the “swiss cheese” model of safety engineering in the paper.
>
> ---
>
> **Q3. Have you tried coding/tool-use benchmarks…**
>
> ---
>
> We see this as a key next step to understanding noise injection more thoroughly, and have taken steps in this direction by including GSM-8K, a free response benchmark for math, in addition to the many multiple choice benchmarks examined.
>
> We intend for future work to verify on both coding and agentic benchmarks.

---

> > ### Comment · Reviewer_QKfV · 2025-08-07
> >
> > Thank you so much for the clarifications. I don't have any further questions.

---

> ### Author Response · Authors · 2025-08-06
>
> Dear Reviewer QKfV,
>
> Thank you for your thoughtful review. We hope our rebuttal has adequately clarified your questions regarding scalability analysis, statistical issues, and the missing p-values. Please let us know if you have any further questions.

---

### Note · Authors · 2025-08-13

We feel the reviewers provided concrete suggestions to improve the internal validity of our work which led to significant improvements to our original submission.

Thanks again to the AC and reviewers for their time and thorough review of our work.

---

### Decision · Program_Chairs · 2025-09-17

**Decision:**

Accept (poster)

**Comment:**

The paper received three expert reviews. The authors provided a rebuttal that attempted to address the concerns raised in the reviews. The reviewers read the rebuttal and engaged with the authors.  The reviewers unanimously like the paper and recommended accept. The area chair agreed with the recommendation and decided to accept the paper. Congratulations! Please see the reviews for feedback on the paper to revise the final version of your paper and include any items promised in your rebuttal.